# Thinking Styles and Creativity: The Mediating Role of Psychological Adjustment in College Students

**DOI:** 10.3390/bs13100875

**Published:** 2023-10-23

**Authors:** Zheng Liu, Huihui Yu, Minxuan Feng, Yubo Hou

**Affiliations:** School of Psychological and Cognitive Sciences, Peking University, Beijing 100871, China; 17710911389@163.com (Z.L.); huihuiyu@stu.pku.edu.cn (H.Y.); minxuan.feng123@stu.pku.edu.cn (M.F.)

**Keywords:** thinking style, holistic thinking, creativity, psychology adjustment, college students

## Abstract

The observation and cultivation of university students’ creativity have always been of enormous concern to the Chinese government. The present research delved into the influence of three dimensions of the Chinese thinking style (Interconnection, Change, and Contradiction) on creativity and the mediating role of psychological adjustment in college students. Specifically, Study 1 investigated the relationship between university students’ thinking styles and creativity through self-reported questionnaires. The results revealed that the thinking style of interconnections had a significant positive effect on creativity among university student groups, while the change dimension significantly negatively affected creativity. The relationship between the dimension of contradiction and creativity did not show significance. Study 2 manipulated thinking style by priming techniques, priming university students’ thinking styles of interconnections (Study 2a) and change (Study 2b), and verified the mediating role of psychological adjustment between thinking styles and creativity. Aside from replicating the findings of Study 1, the results showed that university students’ psychological adjustment mediated the positive effect of interconnections on fluency and the negative effect of change on fluency. These results and findings can provide a theoretical and practical reference for our government to cultivate university students’ creativity.

## 1. Introduction

Societies with high creativity and novelty imply unimaginable potentiality and possibilities. Given the consideration of college students as the core bone of future national development, governments worldwide are committed to improving the creativity of college students. There is no doubt that the Chinese government has also implemented a series of measures to promote creativity in terms of institutional and economic means. However, it is important to recognize that creativity fundamentally revolves around individual behavior. In the endeavor to nurture creative talent, it is critical for college students to understand the subjective psychological factors that drive creativity.

Due to the inherent complexity of creativity itself [1], researchers have often emphasized different aspects when defining creativity, resulting in a lack of a unified and precise definition of creativity to this day. Kozbelt et al. [2] proposed the four P model, which divides the concept of creativity into four dimensions: Creative Product, Creative Cognitive Processes, Creative Person, and Creative Place. When discussing a particular facet of creativity, researchers aim to discern the dynamic interplay between the abilities, processes, and environments of creators. This synergy ultimately leads to the creation of a perceptible product by individuals or groups, one that is both novel and useful within a given social context [3].

Hou [4] indicated that our way of thinking is profoundly influenced by culture. Thinking styles, which represent how individuals process the diverse array of information in their environment, also encompass the way they perceive the world and comprehend their surroundings within cognitive frameworks. In other words, Chinese thinking styles represent a distinctive, lay amalgamation of beliefs, attitudes, and values unique to the Chinese people, which can be activated or inhibited in certain situations [5,6]. Throughout the protracted course of evolution, a growing number of scholars have increasingly regarded thinking styles as enduring metacognitive frameworks [5,7]. Eastern Asian culture, exemplified by China, has unique values and wisdom different from Western culture. Such distinction is especially reflected in the epistemology of dialectical thinking due to the influence of Confucianism, Taoism, Buddhism, and other philosophical thought systems [8]. Meanwhile, the cognitive processes encompassing specific aspects, such as problem-solving patterns, coping mechanisms, and decision-making styles, are intricately interlinked with creativity [3]. Creativity, which can be derived from thinking style [8], is considered a cognitive capacity [9]. Therefore, it is important for psychologists to consider the cultural factor in the relationship between thinking styles and creativity. However, most of the previous studies used tools based on Western cultural characteristics, such as dialectical thinking [10,11,12], that could not fully reflect Chinese thinking styles.

Holistic thinking is an important feature of Chinese culture, including interconnections, change, and contradiction [5]. These thinking styles have great impacts on the process of cognition. Interconnections encompass the recognition of intrinsic interconnections among phenomena and the avoidance of isolating universal relationships. Change requires the examination of issues from the perspective of continuous flux, rather than a state of stagnation. It posits that the essence of the world inherently embodies dynamism, with stasis representing a relative condition. Contradiction underscores the consistent need for adherence to a perspective rooted in contradictions while understanding phenomena. It posits that entities consist of two opposing facets engaged in mutual opposition [4].

For example, Chinese people tend to approach things in the way of interconnections and change. They believe that everything in the world contains contradictions and view compromise as an excellent way to deal with them [13]. Some studies have noted that holistic thinking is not conducive to creativity [14,15,16], especially in the tolerance of contradictions. It suggests that people do not have the motivation to explore and solve contradictions and thus hinder the process of development, integration, and synthesis [14]. Others have claimed that Chinese thinking styles are related to the creativity of steady learning [17] or progressive improvement [18]. However, the above conjecture has not been further verified.

Furthermore, the college phase is a crucial period of individual psychological and physiological maturity, typically occurring within the age bracket of 18 to 22. [19]. In this period, the physiological function of college students is already similar to that of adults, but there is a substantial gap between them in psychological development. Therefore, college students are prone to maladjustment in daily life, such as study and interpersonal relationships, because of their limited life experience and immature mentality in the face of problems. If not handled properly, they may develop psychological problems such as low self-esteem and loneliness [20]. Maladjustment also harms their ability to solve problems and further affects creativity. For example, psychological problems caused by psychological maladaptation directly affect an individual’s cognitive flexibility and even lead to abnormal functions of the frontal cortex, amygdala, and hippocampus in severe cases [21]. Such effects are detrimental to the development of creativity. According to the model of factors influencing creativity, both the creative environment and psychological factors play a significant role in impacting the creativity of college students [22].

Currently, college students are exposed to complex environments, including academic settings, campus life, and interpersonal relationships. Regardless of the environment’s complexity, fostering creativity requires a democratic and harmonious atmosphere. Harmony, in this context, mainly refers to managing and coordinating various relationships. Psychological harmony and social harmony are aligned [22]. Psychological adaptation reflects an individual’s response to their surrounding environment and their own psychological state of harmony. Therefore, high psychological adaptation in college students reflects both their positive response to the surrounding environment and their favorable psychological state, providing a foundation for the enhancement of their creativity [23].

On the other hand, individuals’ level of psychological adaptation is affected by their thinking styles. Hou [24] found that when the thinking style of interconnections is high, individuals tend to adopt positive coping styles, thus improving adaptability. Additionally, individuals with a high level of thinking style of change are more likely to have interpersonal problems [25], and individuals with a high level of thinking style of contradiction have lower adaptability [26]. Therefore, the psychological adaptation of college students may play a mediating role between their thinking styles and creativity.

## 2. Present Research

The present study aimed to explore the influence of college students’ thinking styles on creativity and the mediating role of psychological adaptation. Creativity is not a simple concept that has a single category or cognitive pathway [27,28]. This study focused on creativity, particularly emphasizing the indicator of divergent thinking. Divergent thinking involves the capacity to produce multiple potential answers or diverse solutions for a given problem through fluency, originality, and flexibility [29,30]. Thinking styles pertain to the cognitive approach individuals utilize to perceive problems, rather than serving as a means of problem-solving. On the other hand, creativity represents an approach and method for addressing problems, aiding individuals in effectively managing challenges encountered in life [1]. The cognitive orientation of the Chinese people serves as an adaptation to fundamental mechanisms of living, while psychological adaptation refers to an individual’s established capacity for managing disparities between oneself and the environment. Consequently, we have selected the psychological adaptation of university students as the mediating variable between cognitive style (associative, dialectical, and contextual) and creativity. In essence, the impact of cognitive style on creativity may be mediated through the mechanism of psychological adaptation. Thus, we explored which kind of creativity could be influenced by thinking styles through psychological adaptation. Study 1 explored the relationship between university students’ thinking styles and creativity by questionnaire. Study 2, including two substudies, manipulated thinking style (interconnections or change) by priming techniques to further discover causality. In addition, creativity was measured by fluency, originality, and flexibility in divergent thinking, and university students’ psychological adjustment was considered in the relationship between thinking styles and creativity.

## 3. Study 1

This study aimed to examine the relationship between thinking styles and creativity through self-report questionnaires.

### 3.1. Methods

#### 3.1.1. Participants

A total of 530 college students from China were recruited into this study through the Credamo online platform, and 490 data points (N_male_ = 300, N_female_ = 190, M_age_ = 22, *SD_age_* = 3.42) were retained, excluding the remaining 40 participants who failed the attention test. Participants were apprised of the study’s nature, gave their informed consent, and completed all the aforementioned measures.

#### 3.1.2. Instruments

Thinking styles. The brief version of the questionnaire on Chinese thinking style [5] included 13 items with a 7-point Likert scale (1 = strongly disagree, 7 = strongly agree). The questionnaire focuses on individual ways of understanding the world and consists of three dimensions: interconnections (e.g., many seemingly isolated entities are, in fact, interconnected), change (e.g., each individual possesses a core personality that remains unaltered over time), and contradiction (e.g., I have observed that I frequently engage in actions I do not favor). Therefore, the scores of each subscale are used to represent the participants’ thinking styles. In this study, the internal consistency coefficient of total scales is 0.60 (α_interconnections_ = 0.65, α_change_ = 0.70, α_contradiction_ = 0.71).

Creativity. The originality subscale in the Kirton Adaptation–Innovation Inventory [31] includes 13 items. The scale contains descriptions of an individual’s originality tendency with a 7-point Likert scale (1 = very inconsistent, 7 = very consistent; α = 0.87).

Control variable. Gender, age, and social desirability of college students.

### 3.2. Results

Regression analysis was used to investigate the predictive effect of thinking style on the creativity of college students. We controlled the effect of demographic variables and social desirability to find a model between thinking styles and creativity through stepwise hierarchical multiple regression analyses. The results show that change has a significant negative impact on the creativity of college students (*β* = −0.18, *t* = −4.09, *p* < 0.01), and interconnections have a significant positive effect on creativity (*β* = 0.10, *t* = 2.36, *p* < 0.05), while contradiction has no significant effect on creativity because contradiction did not enter the regression equation (see Table 1).

## 4. Study 2

The present study aimed to explore the causality of the thinking styles and creativity of college students. According to Study 1, there was no significant relationship between contradiction and creativity, so we manipulated thinking styles by priming the concepts of interconnections (Study 2a) and change (Study 2b), leading participants to understand the description of thinking styles. In addition, mediation regression analyses were conducted to examine the mediating role of psychological adaptation between thinking styles and creativity.

## 5. Study 2a

### 5.1. Methods

#### 5.1.1. Participants

A total of 181 college students from China were recruited through the online platform Credamo. Eight college students failed the attention test during the experiment, and 173 subjects were retained (N_male_ = 37, N_female_ = 136, M_age_ = 21.44, *SD_age_* = 2.50). Participants were duly informed about the purpose and procedures of the study, and their informed consent was obtained prior to their engagement in the research.

#### 5.1.2. Instruments

A total of 173 college students were randomly assigned to the experimental group (N = 86) or the control group (N = 87). According to previous studies on cultural construction [32,33], we manipulated interconnections by reading popular science articles. Participants needed to read articles for at least 3 min and write down two or three paragraphs of evidence supporting the above arguments to strengthen the effect of priming. Finally, all participants filled out a scale about how connected they thought things were as a manipulation check. The experimental materials are as follows:

Experimental group: interconnections. A series of recent scientific studies have shown that dialectical thinking is an accurate view of reality. In other words, the basic principle of Chinese dialectics is that nothing is isolated and independent, and everything is connected. Recently, researchers have studied how this thinking style affects people’s beliefs, logic, and cognitive function. For example, a group of academics from the University of Michigan found that people who consider the interconnections of things tend to perform better on problem-solving tasks.

Control group: unconnected. A series of recent scientific studies have shown that linear thinking is an accurate view of reality. In other words, the basic principle of this thinking style is that the existence of things is relatively isolated and independent. Specifically, focusing on the local area and ignoring the background information is more conducive to discovering the essence of things. Recently, researchers have studied how this thinking style affects people’s beliefs, logic, and cognitive function. For example, a group of academics from the University of Michigan found that people who consider the abstract part of things tend to perform better on problem-solving tasks.

In addition, the experimental group and the control group were asked to complete the measurement of psychological adaptation and creativity, respectively.

Psychological adaptation. Using the Chinese university students’ psychological adjustment scale [19], we selected the two highest loaded questions under each subdimension as the short version of the scale, with a total of fourteen items (*α* = 0.82).

Creativity. The Alternative Uses Task (AUT) [34] was selected, which is a classical paradigm for measuring divergent thinking. The task requires participants to write down the nondaily uses of objects over two minutes. In this study, participants were asked to write down creative uses for four objects: slippers, bed sheets, chopsticks, and toothbrushes. Among the 13 objects commonly used in the literature, the four objects most familiar to the participants were screened out through pre-experimental experiments to ensure that the participants could generate creativity when they understood the characteristics of these objects. The AUT was measured before and after the thinking style manipulation, which included three indicators: flexibility, fluency, and originality. Specifically, the pretest scores were used as a control variable in the analysis to isolate the effect of priming. The pretest used slippers and a toothbrush, and the posttest used sheets and chopsticks.

Rating. Fluency scores were calculated based on the number of nonrepeated answers.

Flexibility scores were calculated based on the number of categories in which participants wrote down the uses of an object. For example, slippers can beat people and swat cockroaches, while these uses both belong to the “slap” category. Raters separately assessed the thought flexibility of each participant for each object and finally obtained an average flexibility score (ICC(3) = 0.97, 0.98, 0.97, 0.99).

Originality scores were calculated by three trained raters according to the consensus assessment technique. The rater scored the answers through a 5-point scale (0 = very ordinary, 4 = very novel) and then calculated the average of the top three responses to control for the confusion effect of fluency (ICC(3) = 0.61, 0.58, 0.72, 0.69; [35]).

Control variables. Gender, age.

### 5.2. Results

Manipulation check. The analysis of covariance (ANCOVA) tested the effectiveness of initiating the interconnections. After controlling for gender and age, the experimental group (M = 6.16, *SD* = 0.59) scored significantly higher on the test questions than the control group (M = 5.79, *SD* = 0.63), *F* (1, 169) = 15.16, *p* < 0.001, $\eta^{2}$ = 0.08. Therefore, the priming effect is effective.

#### 5.2.1. The Mediating Role of Psychological Adaptation between Interconnections and Fluency

A multilevel regression method was used to examine the mediating role of psychological adaptation between interconnections and fluency. The results, as shown in Table 2, revealed that interconnections significantly positively predicted fluency, *β* = 0.37, *p* < 0.001, 95% CI = [0.17, 0.58]. At the same time, interconnections positively predicted college students’ psychological adaptation, *β* = 0.55, *p* < 0.001, 95% CI = [0.27, 0.82]. After controlling for gender, age, pretest fluency, and interconnections, psychological adaptation significantly positively predicted posttest fluency, *β* = 0.18, *p* < 0.01, 95% CI = [0.07, 0.29].

We used Model 4 in Hayes’s process to calculate the mediating role of the psychological adaptation between interconnections and fluency using SPSS 26.0 with PROCESS v3.3. Gender, age, and pretest fluency were used as control variables, interconnections as independent variables (1 = experimental group, 0 = control group), posttest fluency as a dependent variable, and psychological adaptation as a mediating variable. The results showed that the 95% confidence interval for the standardized indirect effect obtained from 5000 bootstrap tests was [0.03, 0.18], excluding 0, suggesting that psychological adaptation plays a mediating role in the effect of interconnections on fluency (see Figure 1).

#### 5.2.2. The Mediating Role of Psychological Adaptation between Interconnections and Flexibility

A multilevel regression method was used to examine the mediating role of psychological adaptation between interconnections and flexibility. Interconnections significantly positively predicted posttest flexibility, *β* = 0.23, *p* < 0.05, 95% CI = [0.01, 0.45]; interconnections significantly positively predicted psychological adaptation, *β* = 0.54, *p* < 0.001, 95% CI = [0.27, 0.82]; after controlling for gender, age, pretest flexibility, and interconnections, psychological adaptation did not significantly predict posttest flexibility, *β* = 0.03, *p* = 0.66 (see Table 3).

We used Model 4 in Hayes’s process to calculate the mediating role of the psychological adaptation between interconnections and fluency using SPSS 26.0 with PROCESS v3.3. Gender, age, and pretest flexibility were used as control variables, interconnections as independent variables (1 = experimental group, 0 = control group), posttest flexibility as the dependent variable, and psychological adaptation as a mediating variable. The results showed that the 95% confidence interval for the standardized indirect effect obtained from 5000 bootstrap tests was [−0.07, 0.09], including 0, indicating that psychological adaptation has no mediating effect of interconnections on flexibility.

#### 5.2.3. The Mediating Role of Psychological Adaptation between Interconnections and Originality

A multilevel regression method was used to examine the mediating role of psychological adaptation between interconnections and originality. Interconnections did not significantly predict posttest originality, *β* = 0.01, *p* = 0.96; interconnections significantly positively predicted psychological adaptation, *β* = 0.56, *p* < 0.001, 95% CI = [0.28, 0.83]; after controlling for gender, age, pretest originality, and interconnections, psychological adaptation did not significantly predict posttest originality, *β* = −0.09, *p* = 0.25 (see Table 4).

We used Model 4 in Hayes’s process to calculate the mediating role of the psychological adaptation between interconnections and fluency using SPSS 26.0 with PROCESS v3.3. Gender, age, and pretest originality were used as control variables, interconnections as independent variables (1 = experimental group, 0 = control group), posttest originality as the dependent variable, and psychological adaptation as a mediating variable. The results showed that the 95% confidence interval for the standardized indirect effect obtained from 5000 bootstrap tests was [−0.14, 0.02], containing 0, indicating that psychological adaptation has no mediating effect of interconnections on originality.

## 6. Study 2b

### 6.1. Methods

#### 6.1.1. Participants

Through the online platform Credamo, a total of 180 college students from China were recruited. Nine college students failed the attention test during the experiment, and 171 subjects were retained (N_male_ = 39, N_female_ = 132, M_age_ = 21.25, *SD_age_* = 3.08). Participants were provided with a comprehensive overview of the study’s objectives, methodology, and procedures, ensuring their informed consent was obtained before their active participation in the research.

#### 6.1.2. Instruments

A total of 171 college students were randomly assigned to the experimental group (N = 86) or the control group (N = 87). First, according to previous studies on cultural construction [32,33], we manipulated change by reading popular science articles. Participants needed to read articles for at least 3 min and write down two or three paragraphs of evidence supporting the above arguments to strengthen the effect of priming. Finally, all participants filled out a scale about how much they thought things were constantly changing as a manipulation check. The experimental materials are as follows:

Experimental group: change. A series of recent scientific studies have shown that dialectical thinking is an accurate view of reality. In other words, the basic principle of Chinese dialectics is that everything is constantly changing because reality is a process. Recently, researchers have studied how this thinking style affects people’s beliefs, logic, and cognitive function. For example, a group of academics from the University of Michigan found that people who consider the change of things tend to perform better on problem-solving tasks.

Control group: unchanged. A series of recent scientific studies have shown that linear thinking is an accurate view of reality. In other words, the basic principle of linear thinking is that the world is stable and consistent, and truth must exist. Therefore, analysis and linear thinking are useful. Recently, researchers have studied how this thinking style affects people’s beliefs, logic, and cognitive function. For example, a group of academics from the University of Michigan found that people who used analysis and reasoning to approach answers tend to perform better on problem-solving tasks.

In addition, the experimental group and the control group were asked to complete the measurement of psychological adaptation and creativity, respectively. The measurement of psychological adaptation (*α* = 0.83) and creativity was consistent with that in Study 2a. Specifically, the interrater reliability of originality and flexibility was good (ICC(3)_originality_ = 0.67, 0.61, 0.74, 0.65; ICC(3)_flexibility_ = 0.94, 0.95, 0.98, 0.99).

Control variables. Gender, age.

### 6.2. Results

Manipulation check. The analysis of covariance (ANCOVA) tested the effectiveness of initiating change. After controlling for gender and age, the experimental group (M = 6.29, *SD* = 0.85) scored significantly higher on the test questions than the control group (M = 5.56, *SD* = 0.80), *F* (1, 167) = 23.40, *p* < 0.001, $\eta^{2}$ = 0.12. Therefore, the priming effect is effective.

#### 6.2.1. The Mediating Role of Psychological Adaptation between Change and Fluency

A multilevel regression method was used to examine the mediating role of psychological adaptation between change and fluency. The results indicated that change significantly negatively predicted fluency, *β* = −0.36, *p* < 0.01, 95% CI = [−0.59, −0.14]. At the same time, change negatively predicted psychological adaptation, *β* = −0.37, *p* < 0.05, 95% CI = [−0.67, −0.07]. After controlling for gender, age, pretest fluency, and change, psychological adaptation significantly positively predicted posttest fluency, *β* = 0.18, *p* < 0.01, 95% CI = [0.06, 0.29] (see Table 5).

We used Model 4 in Hayes’s process to calculate the mediating role of psychological adaptation between change and fluency using SPSS 26.0 with PROCESS v3.3. Gender, age, and pretest fluency were used as control variables, change as independent variables (1 = experimental group, 0 = control group), posttest fluency as the dependent variable, and psychological adaptation as a mediating variable. The results showed that the 95% confidence interval for the standardized indirect effect obtained from 5000 bootstrap tests was [−0.14, −0.01], excluding 0, suggesting that psychological adaptation plays a mediating role in the effect of change on fluency (see Figure 2).

#### 6.2.2. The Mediating Role of Psychological Adaptation between Change and Flexibility

A multilevel regression method was used to examine the mediating role of psychological adaptation between change and flexibility. Change significantly negatively predicts posttest flexibility, *β* = −0.26, *p* < 0.05, 95% CI = [−0.51, −0.003]; change significantly negatively predicts psychological adaptation, *β* = −0.38, *p* < 0.05, 95% CI = [−0.68, −0.08]; after controlling for gender, age, pretest flexibility, and change, psychological adaptation does not significantly predict posttest flexibility, *β* = −0.05, *p* = 0.44 (see Table 6).

We used Model 4 in Hayes’s process to calculate the mediating role of psychological adaptation between change and fluency using SPSS 26.0 with PROCESS v3.3. Gender, age, and pretest flexibility were used as control variables, change was used as an independent variable (1 = experimental group, 0 = control group), posttest flexibility was used as the dependent variable, and psychological adaptation was used as a mediating variable. The results showed that the 95% confidence interval for the standardized indirect effect obtained from 5000 bootstrap tests was [−0.03, 0.08], including 0, indicating that psychological adaptation has no mediating effect of change on flexibility.

#### 6.2.3. The Mediating Role of Psychological Adaptation between Change and Originality

A multilevel regression method was used to examine the mediating role of psychological adaptation between change and originality. Change did not significantly predict posttest originality, *β* = −0.10, *p* = 0.46; change significantly negatively predicted psychological adaptation, *β* = −0.37, *p* < 0.05, 95% CI = [−0.67, −0.07]; after controlling for gender, age, pretest originality, and change, psychological adaptation did not significantly predict posttest originality, *β* = −0.02, *p* = 0.77 (see Table 7).

We used Model 4 in Hayes’s process to calculate the mediating role of psychological adaptation between change and fluency using SPSS 26.0 with PROCESS v3.3. Gender, age, and pretest originality were used as control variables, change was used as an independent variable (1 = experimental group, 0 = control group), posttest originality was used as the dependent variable, and psychological adaptation was used as a mediating variable. The results showed that the 95% confidence interval for the standardized indirect effect obtained from 5000 bootstrap tests was [−0.05, 0.07], including 0, indicating that psychological adaptation has no mediating effect of change on originality.

## 7. Discussion

The present study explored the influence of college students’ thinking styles on creativity and the mediating role of psychological adaptation. The results show that interconnections have a significant positive impact on creativity, change has a significant negative impact on creativity, and contradiction has no significant impact on creativity (Study 1). Interconnections reflects an individual’s tendency to discover and establish links between different concepts. Individuals with a strong inclination for making connections tend to be more proficient at extrapolating instances from one scenario to another and deducing from existing information, leading to novel discoveries. Additionally, researchers have explored the relationship between cognitive styles and creativity, adopting a holistic thinking approach that is prevalent among Chinese individuals. For example, Hou et al. [36] studied creativity among Chinese employees in organizations and found that a propensity for connective thinking effectively predicted an individual’s level of creativity. Specifically, individuals with a higher propensity for connective thinking can integrate and utilize existing information, thus enhancing their creative potential. Conversely, a disposition toward variable thinking could potentially diminish the creative capabilities of university students. According to Hou et al. [5], individuals with lower variability scores are more likely to express their thoughts in the presence of external disruptions. They also tend to display greater confidence in their attributes when addressing problems. In contrast, individuals with a higher propensity for variability often exhibit reduced stability and consistency in their perceptions, resulting from uncertainty and a lack of steadfast beliefs in certain traits. Hou et al. [36] further discovered that individuals with a higher propensity for variability, due to their inherent uncertainty and difficulty in steadfastly adhering to unchanging characteristics, might become prone to distraction, leading to a decline in creativity. During the university phase, cognitive styles have yet to fully stabilize. With higher levels of variability in thinking, students may exhibit interest in a wide array of elements within their environment, resulting in reduced concentration and a persistent drive to explore novel aspects. This propensity can, in turn, contribute to a decline in their creative capabilities.

Furthermore, the findings from Study 2 demonstrated that thinking styles mediated the relationship between the fluency dimension of creativity and psychological adaptation. Specifically, there is a positive correlation between interconnections and creativity, whereas there is a negative correlation between change and creativity. More specifically, in relation to various creativity indexes, interconnections demonstrate a positive association with the fluency and flexibility of divergent thinking. Conversely, change exhibits a negative association with the fluency and flexibility of divergent thinking. Additionally, it is noteworthy that psychological adaptation serves as a mediating factor in the relationship between interconnections and fluency. Interconnections exert a positive influence on psychological adaptation, subsequently enhancing the quantity of creative ideas (fluency). Similarly, psychological adaptation also mediates the relationship between change and fluency, as change exerts a negative impact on psychological adaptation, resulting in a reduction in the quantity of creative ideas (fluency).

### 7.1. Implication

Interconnections, a network structure of individual thinking, reflect the individual’s holistic view of things [4]. People with a high level of interconnections tend to believe that everything is connected and that nothing exists in isolation. Therefore, interconnections prompt people to look for an association between things, which can help people understand the rest by analogy [36]. Creativity requires us to develop something new by systematically integrating a set of relevant knowledge [37]. It follows that the key to creativity is the ability to make interconnections between things that others thought were unconnected and to open up new paths that no one else thought of.

Change, as one of the basic beliefs of Chinese people looking at the world, has had a profound impact on the development and inheritance of thought in Chinese history [38]. Western traditional thinking styles are usually regarded as a kind of linear thinking, while Chinese traditional thinking styles are a kind of nonlinear thinking [10]. In other words, Chinese people believe that reality is a dynamic process [8] and tend to consider changes in things more than Westerners due to differences in thinking styles. On the third day of their separation, they should look at each other with new eyes. However, there is a certain contradiction between change and the basic requirement of creativity, such as divergent thinking [39]. The thinking style of change infers that things are in constant reciprocating processes between opposite elements with a temporal balancing [40,41], which may inhibit participants’ thinking about the possibility of more uses for objects in the measurement of creativity because participants would dwell on the interaction and interconversion between opposite elements. In this case, change is likely to interfere with the process of pursuing innovation and, to some extent, inhibit the exploration and persistence of new and different ideas.

Because of the fundamental cognitive differences in the contradiction between Eastern and Western cultures, Western researchers have always believed that Hegel’s dialectical thinking is more conducive to the improvement of creativity than naïve dialectical thinking [10,42,43]. Hegel’s dialectical thinking shows the innovative path of “contradiction emergence—tension of opposition—problem-solving—innovative solutions”. That is, the opposition or conflict between elements mediates the positive impact of contradiction on creativity [10]. Ward and colleagues [44] pointed out that higher levels of innovation (breakthrough creativity) tend to be achieved when trying to solve the opposite relationship between different elements. In contrast, under the influence of naïve dialectical thinking, Easterners are less stimulated by the opposition and seek solutions to contradictions. In other words, Easterners are more likely to accept the tension of opposition and seek an “intermediate” solution [45]. Therefore, Western dialectical thinking emphasizes the creation of a new solution to contradictions through integration or synthesis, while Eastern dialectical thinking regards contradictions as objective existence and chooses to tolerate and accept them. In the present study, there is no significant relationship between contradictions and creativity. The possible reason might be that a high tolerance for contradictions, on the one hand, facilitates the combination of opposing elements and, on the other hand, has certain difficulties in generating new ideas immediately. Therefore, naïve dialectical thinking is likely associated with steady learning [17] or progressive improvement [18].

Psychological adaptation refers to the ability of individuals to change themselves or the environment in the process of socialization and finally enable themselves to live in harmony with the environment [46,47]. When facing problems, people with a low level of psychological adaptation perceive a high level of aloneness [48] and show more aggression [49]. However, people with a high level of psychological adaptation regulate their emotions well [50]. Furthermore, mental health provides a good foundation for creativity [21,51]. In terms of motivation, most individuals’ needs are satisfied with creative activities or lead to maladjustment and even illness if not met [52]. College students, to adapt to difficulties, have the motivation to exert more creativity and obtain the corresponding psychological resources to help them solve problems, especially those with a high level of psychological adaptation. Hence, psychological adaptation helps individuals save cognitive resources and create more possibilities for creativity.

### 7.2. Limitations and Future Research

First, we did not find a relationship between college students’ contradictions and creativity in Study 1, so we did not continue to induce thinking contradictions in the follow-up Study 2. Future research needs to further explore this issue by inducing thinking contradictions.

Second, the raters, master’s students from key universities in Study 2, were trained, but they also gave low scores to the participants’ creative answers due to their high levels of intelligence and creativity.

Third, because the study was performed during COVID-19, many participants could not come to the laboratory. Hence, most of our research was conducted online. Normally, the standard duration of AUT is 4 min [53], which can ensure that subjects have enough time to form deeper associations. However, since the experiment was moved from an offline to an online platform, we set the time for subjects to 2 min to ensure that subjects had a more suitable and comfortable answering time environment. More cautious laboratory studies are warranted in the future.

Fourth, due to disparities in Eastern and Western cultures, Western individuals tend to favor linear thinking over holistic thinking [38,54]. The divergence in how Chinese individuals’ thinking processes, characterized by three dimensions encompassing connection, change, and contradiction, influence creativity in contrast to their Western counterparts warrants further investigation in future research.

## 8. Conclusions

In the present study, the results suggested that the influence of interconnections and change on creativity is mainly reflected in fluency and flexibility, which indicated that Chinese college students tend to pay more attention to the number of creative ideas, the breadth of thinking, and the ability to solve creative remote associative tasks (convergent thinking) but are not good at originality. In addition, psychological adaptation plays a vital role in college students’ thinking styles and creativity. Explicitly, we explored the effect of thinking styles’ priming on creativity, especially in fluency, through the mediating role of psychological adaptation.

## Figures and Tables

**Figure 1 behavsci-13-00875-f001:**
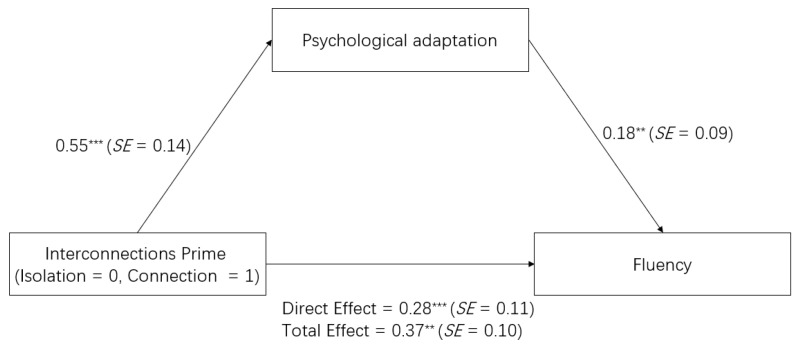
Mediating Effect of Psychological Adaptation on the Relationship Between Interconnections and Fluency in College Students. Note. ** *p* < 0.01. *** *p* < 0.001.

**Figure 2 behavsci-13-00875-f002:**
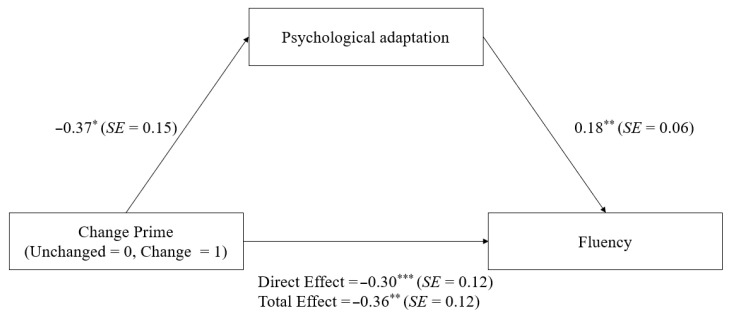
Mediating Effect of Psychological Adaptation on the Relationship Between Change and Fluency in College Students. Note. * *p* < 0.05. ** *p* < 0.01. *** *p* < 0.001.

**Table 1 behavsci-13-00875-t001:** Influence of College Students’ Thinking Style on Creativity.

Model Prediction	B	*SE*	*β*	*t*	*p*
1 Social Desirability	1.02	0.15	0.34	6.88	0.00
Age	0.11	0.13	0.04	0.81	0.42
Gender	−1.76	0.89	−0.09	−1.98	0.05
2 Social Desirability	0.94	0.15	0.32	6.48	0.00
Age	0.06	0.13	0.02	0.43	0.67
Gender	−1.5	0.88	−0.08	−1.71	0.09
Change	−0.45	0.11	−0.19	−4.28	0.00
3 Social Desirability	0.90	0.15	0.31	6.11	0.00
Age	0.04	0.13	0.02	0.34	0.73
Gender	−1.35	0.87	−0.07	−1.55	0.12
Change	−0.43	0.11	−0.18	−4.09	0.00
Interconnections	0.36	0.15	0.10	2.36	0.02

**Table 2 behavsci-13-00875-t002:** Mediating Effect of Psychological Adaptation on the Relationship Between Interconnections and Fluency in College Students.

Predictive Variable	Pretest Fluency		Psychological Adaptation		Protest Fluency	
	*β (SE)*	*t*	*β (SE)*	*t*	*β (SE)*	*t*
Age	0.05 (0.06)	0.97	0.21 (0.07)	2.95 **	0.01 (0.07)	0.26
Gender	−0.08 (0.06)	−1.51	0.21 (0.06)	3.04 **	−0.12 (0.07)	−2.24 *
Pretest Fluency	0.72 (0.06)	14.03 ***	0.13 (0.07)	1.92	0.70 (0.08)	13.78 ***
Interconnections Prime	0.37 (0.10)	3.63 ***	0.55 (0.14)	3.93 ***	0.28 (0.11)	2.65 **
Psychological Adaptation					0.18 (0.09)	3.20 **
R2	0.55		0.19		0.58	
F	52.15 ***		9.77 ***		46.05 ***	

Note. Interconnections Prime (0 = control group; 1 = experimental group). * *p* < 0.05. ** *p* < 0.01. *** *p* < 0.001.

**Table 3 behavsci-13-00875-t003:** Mediating Effect of Psychological Adaptation on the Relationship Between Interconnections and Flexibility in College Students.

Predictive Variable	Pretest Flexibility		Psychological Adaptation		Protest Flexibility	
	*β (SE)*	*t*	*β (SE)*	*t*	*β (SE)*	*t*
Age	0.04 (0.06)	0.78	0.20 (0.07)	2.93 **	0.04 (0.06)	0.67
Gender	−0.10 (0.06)	−1.82	0.21 (0.07)	3.06 **	−0.11 (0.06)	−1.87
Pretest Flexibility	0.67 (0.06)	11.89 ***	0.11 (0.07)	1.59	0.67 (0.06)	11.72 ***
Interconnections Prime	0.23 (0.11)	2.03 *	0.54 (0.14)	3.89 ***	0.21 (0.11)	1.81
Psychological Adaptation					0.03 (0.06)	0.45
R2	0.47		0.18		0.47	
F	37.01 ***		9.43 ***		29.51 ***	

Note. Interconnections Prime (0 = control group; 1 = experimental group). * *p* < 0.05. ** *p* < 0.01. *** *p* < 0.001.

**Table 4 behavsci-13-00875-t004:** Mediating Effect of Psychological Adaptation on the Relationship Between Interconnections and Originality in College Students.

Predictive Variable	Pretest Originality		Psychological Adaptation		Protest Originality	
	*β (SE)*	*t*	*β (SE)*	*t*	*β (SE)*	*t*
Age	−0.06 (0.07)	−0.88	0.20 (0.07)	2.84 **	−0.04 (0.07)	−0.61
Gender	−0.08 (0.07)	−1.09	0.21 (0.07)	3.02 **	−0.06 (0.07)	−0.80
Pretest Originality	0.38 (0.07)	5.31 ***	0.04 (0.07)	0.51	0.38 (0.07)	5.35 ***
Interconnections Prime	0.01 (0.07)	0.04	0.56 (0.07)	3.94 ***	0.06 (0.07)	0.38
Psychological Adaptation					−0.09 (0.08)	0.45
R2	0.16		0.17		0.17	
F	8.14 ***		8.78 ***		6.79 ***	

Note. Interconnections Prime (0 = control group; 1 = experimental group). ** *p* < 0.01. *** *p* < 0.001.

**Table 5 behavsci-13-00875-t005:** Mediating Effect of Psychological Adaptation on the Relationship Between Change and Fluency in College Students.

Predictive Variable	Pretest Fluency		Psychological Adaptation		Protest Fluency	
	*β(SE)*	*t*	*β(SE)*	*t*	*β(SE)*	*t*
Age	0.13 (0.06)	2.15 *	0.01 (0.08)	0.09	0.13 (0.06)	2.19 *
Gender	0.05 (0.06)	0.78	−0.04 (0.08)	−0.55	0.05 (0.06)	0.92
Pretest Fluency	0.59 (0.06)	9.93 ***	0.10 (0.08)	1.24	0.57 (0.06)	9.84 ***
Change Prime	−0.36 (0.12)	−3.08 **	−0.37 (0.15)	−2.40 *	−0.30 (0.12)	−2.55 ***
Psychological Adaptation					0.18 (0.06)	3.03 **
R2	0.44		0.05		0.47	
F	32.86 ***		2.18		29.42 ***	

Note. Change Prime (0 = control group; 1 = experimental group). * *p* < 0.05. ** *p* < 0.01. *** *p* < 0.001.

**Table 6 behavsci-13-00875-t006:** Mediating Effect of Psychological Adaptation on the Relationship Between Change and Flexibility in College Students.

Predictive Variable	Pretest Flexibility		Psychological Adaptation		Protest Flexibility	
	*β (SE)*	*t*	*β (SE)*	*t*	*β (SE)*	*t*
Age	0.04 (0.06)	0.60	−0.00 (0.08)	−0.03	0.04 (0.06)	0.55
Gender	0.06 (0.06)	0.97	−0.04 (0.08)	−0.53	0.06 (0.06)	0.94
Pretest Flexibility	0.54 (0.06)	8.32 ***	0.13 (0.08)	1.67	0.55 (0.06)	8.34 ***
Change Prime	−0.26 (0.12)	−1.99 *	−0.38 (0.15)	−2.48 *	−0.28 (0.12)	−2.10 *
Psychological Adaptation					−0.05 (0.06)	−0.78
R2	0.33		0.06		0.33	
F	20.47 ***		2.51 *		16.46 ***	

Note. Change Prime (0 = control group; 1 = experimental group). * *p* < 0.05. *** *p* < 0.001.

**Table 7 behavsci-13-00875-t007:** Mediating Effect of Psychological Adaptation on the Relationship Between Change and Originality in College Students.

Predictive Variable	Pretest Originality		Psychological Adaptation		Protest Originality	
	*β (SE)*	*t*	*β (SE)*	*t*	*β (SE)*	*t*
Age	0.08 (0.07)	1.20	0.01 (0.08)	0.08	0.08 (0.07)	1.19
Gender	0.14 (0.07)	2.03 *	–0.02 (0.08)	−0.30	0.14 (0.07)	2.01 *
Pretest Originality	0.42 (0.07)	5.99 ***	0.08 (0.08)	1.05	0.42 (0.07)	5.97 ***
Change Prime	−0.10 (0.14)	−0.74	−0.37 (0.15)	−2.42 *	−0.11 (0.14)	−0.78
Psychological Adaptation					−0.02 (0.07)	−0.29
R2	0.21		0.05		0.21	
F	11.30 ***		2.07		9.01 ***	

Note. Change Prime (0 = control group; 1 = experimental group). * *p* < 0.05. *** *p* < 0.001.

## Data Availability

The data are available from the corresponding author on reasonable request.

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
