# Peer review of "Thinking Styles and Creativity: The Mediating Role of Psychological Adjustment in College Students"

_behavsci, 2023, doi:10.3390/bs13100875_

Round 1

Reviewer 1 Report

Introduction

 ++ Overall, the introduction is not very comprehensive. Border research  framework and motivation for the current study is not fully explained. What is the novelty of the present research and its theoretical contribution?

 ++ It would be very helpful to see the definitions of different types of cognitive styles discussed in the paper. For example, thinking style of change or thinking style of contradiction. Creativity is also rather a broad term, and its more specific focused  explanation about different dimensions of creativity will help. The statement: ‘Meanwhile, creativity is not a simple concept’ comes too late, I would recommend that the explanation about the constructs of interest should be part on intro, before talking about the specific study.

 pp.58-59. ‘psychological maladjustment can lead to negative emotions [14] and even reduce self-efficacy [15], thereby inhibiting the development of creativity.’

 ++ Is there opposite evidence as well? I am not very familiar with the literature, but  there are so many examples of very successful artists and writers who suffered depression and negative emotions.

Study 1

Sample size is impressive.  

Three dimensions: interconnections, change and contradiction should be defined,

Including some examples of items would be helpful.

The results do not include the discussions of all the significant values, and Table 1 does not include contradiction. Even if there is no significant effect, the full results should be presented.

Study 2

It’s not clear enough how the given tasks manipulated interconnections or change by reading popular science articles. Again, the definition of interconnections and change and the link between how they are operationalized in the first and second study would help.

Discussion

pp. 356-357 ‘thinking styles mediated the relationship between the fluency dimension of creativity and psychological adaptation’. Please, elaborate this statement.

Researchers discuss their findings in the light of cognitive differences between Eastern and Western cultures; however, present research is not based on cross-cultural comparison. Would the findings generalize to other cultures? How do the current findings contribute to the existing literature beyond Chinese sample?

Some statements (pp.384-385) ‘Western researchers have always believed that Hegel’s dialectical thinking is more conducive to the improvement of creativity than naïve dialectical thinking’ sound like oversimplifications.

Author Response

Dear Reviewer,

I would like to express my sincere gratitude for your invaluable guidance. Below, you will find the revisions I have made based on your insightful recommendations. Your feedback has been immensely helpful, and I am deeply appreciative of your support and expertise.

Introduction

 ++ Overall, the introduction is not very comprehensive. Border research  framework and motivation for the current study is not fully explained. What is the novelty of the present research and its theoretical contribution?

Response: Thanks for your advice. We have revised the introduction on page 1-2.

++ It would be very helpful to see the definitions of different types of cognitive styles discussed in the paper. For example, thinking style of change or thinking style of contradiction. Creativity is also rather a broad term, and its more specific focused  explanation about different dimensions of creativity will help. The statement: ‘Meanwhile, creativity is not a simple concept’ comes too late, I would recommend that the explanation about the constructs of interest should be part on intro, before talking about the specific study.

Response: I deeply appreciate your valuable suggestion. In response, I have incorporated more specific conceptual definitions into the introduction.

pp.58-59. ‘psychological maladjustment can lead to negative emotions [14] and even reduce self-efficacy [15], thereby inhibiting the development of creativity.’

 ++ Is there opposite evidence as well? I am not very familiar with the literature, but  there are so many examples of very successful artists and writers who suffered depression and negative emotions.

Response: I humbly express my gratitude for your mentorship. In this manuscript, my objective is to clarify that when individuals experience prolonged psychological maladjustment, the continuity of creativity can be hindered by negative emotions. This is due to the fact that negative emotions can have a detrimental impact on cognitive processes, whereas creativity requires a clear understanding and definition of problems, along with the ability to prepare resources for addressing them.

Study 1

Sample size is impressive.  

Three dimensions: interconnections, change and contradiction should be defined,

Including some examples of items would be helpful.

The results do not include the discussions of all the significant values, and Table 1 does not include contradiction. Even if there is no significant effect, the full results should be presented.

Response: Because the value of contradiction is not significant, we did not include it according to academic requirements.

Study 2

It’s not clear enough how the given tasks manipulated interconnections or change by reading popular science articles. Again, the definition of interconnections and change and the link between how they are operationalized in the first and second study would help.

Response: I sincerely appreciate your valuable suggestion. In response, I have incorporated a detailed exposition of the three dimensions of Chinese-style thinking in the introduction. This addition is aimed at providing a clearer explanation of the manipulation in our study.

Discussion

356-357 ‘thinking styles mediated the relationship between the fluency dimension of creativity and psychological adaptation’. Please, elaborate this statement.

Response: Thank you for your valuable suggestion. I have added the results of each study in the first paragraph of the discussion. Your input is greatly appreciated.

Researchers discuss their findings in the light of cognitive differences between Eastern and Western cultures; however, present research is not based on cross-cultural comparison. Would the findings generalize to other cultures? How do the current findings contribute to the existing literature beyond Chinese sample?

Response: I deeply appreciate your valuable feedback. This study primarily focuses on the thinking styles of Chinese university students. Consequently, the research did not address the differences in these three thinking styles among Western university students. I will make sure to address this gap in our research.

Some statements (pp.384-385) ‘Western researchers have always believed that Hegel’s dialectical thinking is more conducive to the improvement of creativity than naïve dialectical thinking’ sound like oversimplifications.

Response: I sincerely appreciate your valuable suggestion. In response to it, I have incorporated citations into the discussion section. Furthermore, I have also included the perspective of scholars on Hegel's dualistic thinking advantage.

I wish to once again convey my heartfelt gratitude and eagerly await your response.

Sincerely,

Huihui Yu.

Reviewer 2 Report

I appreciate the authors for their work. I believe that the authors have invested substantial effort into the methodological aspect of the manuscript, particularly in the statistical analyses. Below, I present a series of minor comments/observations/suggestions that I hope will contribute to enhancing your work.

Lines 34-36, this argument mentions that previous studies have used tools based on Western culture; however, the argument is only supported by one reference/study.

Lines 41-44, this argument mentions that some studies have observed that holistic thinking is not conducive to creativity; however, the argument is only supported by a single reference/study.

Analysis and Results Section

As a suggestion for the analyses conducted in the various studies:

In the regression analysis, have you observed any issues with collinearity? You could perform the Variance Inflation Factor (VIF) to ensure accuracy. Another approach is to conduct a Principal Component Analysis and a Correlation Biplot (with lambda equal to 0) to check for correlations among the independent variables. This is just a suggestion...

Have you attempted Differential Item Functioning (DIF) tests using methods such as Lord, Raju, or Mantel-Haenszel (MH)? This analysis could be interesting to understand if there's something particular happening with your sample or if it's related to the questionnaire. While not obligatory, I believe it could add more depth, but what you've presented so far is fine. I also recommend taking a look at it.

Lines 394-396 Interesting...

Lines 432-435 Interesting...

References. Only one reference out of 39 is from the year 2020 onwards. It would be beneficial to include more current references to enhance the content.

Author Response

Dear Reviewer,

I would like to express my sincere gratitude for your invaluable guidance. Below, you will find the revisions I have made based on your insightful recommendations. Your feedback has been immensely helpful, and I am deeply appreciative of your support and expertise.

Lines 34-36, this argument mentions that previous studies have used tools based on Western culture; however, the argument is only supported by one reference/study.

Response: I wish to express my profound gratitude for your kind suggestions, which have greatly enriched the content by enabling the inclusion of additional scholarly references.

Lines 41-44, this argument mentions that some studies have observed that holistic thinking is not conducive to creativity; however, the argument is only supported by a single reference/study.

Response: Your guidance has been invaluable, and I want to extend my heartfelt appreciation for your recommendations, which have enabled me to include additional relevant literature.

Analysis and Results Section

As a suggestion for the analyses conducted in the various studies:

In the regression analysis, have you observed any issues with collinearity? You could perform the Variance Inflation Factor (VIF) to ensure accuracy. Another approach is to conduct a Principal Component Analysis and a Correlation Biplot (with lambda equal to 0) to check for correlations among the independent variables. This is just a suggestion...

Response:

We have performed the Variance Inflation Factor (VIF) to ensure no collinearity of the model.

Have you attempted Differential Item Functioning (DIF) tests using methods such as Lord, Raju, or Mantel-Haenszel (MH)? This analysis could be interesting to understand if there's something particular happening with your sample or if it's related to the questionnaire. While not obligatory, I believe it could add more depth, but what you've presented so far is fine. I also recommend taking a look at it.

Response: Thank you for your recommendation. Differential Item Functioning (DIF) tests are commonly utilized in studies involving subgroups or cross-cultural research. However, in the context of this study, where our sample consists solely of university students, this method may not be suitable. Nevertheless, we acknowledge your suggestion, and we intend to incorporate this approach in future cross-cultural research studies. We greatly value your guidance

Lines 394-396 Interesting...

Lines 432-435 Interesting...

References. Only one reference out of 39 is from the year 2020 onwards. It would be beneficial to include more current references to enhance the content.

Response: I wish to express my profound gratitude for your kind suggestions, which have greatly enriched the content by enabling the inclusion of additional scholarly references.

I wish to once again convey my heartfelt gratitude and eagerly await your response.

Sincerely,

Huihui Yu.

Reviewer 3 Report

Thank you very much for the opportunity to read and revise this paper on the mediating effect of psychological adjustment in the association between thinking styles and creativity. 

Overall, the paper is well-written and explores an interesting topic concerning the mechanisms underpinning creativity in college students. Despite some strengths, the manuscript shows some problems, especially from a theoretical point of view.

Below are my comments and suggestions, which, I hope, could help improve the quality of this work. 

1) The authors based their work on creativity. Nevertheless, I noted that no description and definition of creativity has been provided. I suggest authors include a clear conceptualisation of creativity and its main features. The papers below could be helpful for this aim: 

- Runco, M. A., & Jaeger, G. J. (2012). The standard definition of creativity. Creativity Research Journal, 24(1), 92-96.

- Giancola, M., Palmiero, M., Piccardi, L., & D’Amico, S. (2022). The relationships between cognitive styles and creativity: The role of field dependence-independence on visual creative production. Behavioral Sciences12(7), 212.

Additionally, given that the authors employed the alternative uses task, a clear description of divergent thinking is necessary. The papers below could be useful for this aim:

- Acar, S., & Runco, M. A. (2019). Divergent thinking: New methods, recent research, and extended theory. Psychology of Aesthetics, Creativity, and the Arts, 13(2), 153.

- Giancola, M., Palmiero, M., Bocchi, A., Piccardi, L., Nori, R., & D’Amico, S. (2022). Divergent thinking in Italian elementary school children: The key role of probabilistic reasoning style. Cognitive Processing, 23(4), 637-645.

2) The concept of thinking style seems to come out of nowhere. I understand that culture greatly affects people’s way of thinking, but what does represent thinking style?

I advise authors to describe thinking style and then explain the influence of culture. Overall, this description should also be useful for introducing holistic thinking. 

The papers reported below should be useful: 

- Li, L. M. W., Masuda, T., Hamamura, T., & Ishii, K. (2018). Culture and decision making: Influence of analytic versus holistic thinking style on resource allocation in a fort game. Journal of Cross-Cultural Psychology49(7), 1066-1080. 

- Cheng, H., Andrade, H. L., & Yan, Z. (2011). A cross-cultural study of learning behaviours in the classroom: From a thinking style perspective. Educational Psychology31(7), 825-841.

Also, I suggest authors explain thinking style theoretically, reporting a well-established model, allowing the reader to understand the concept of thinking style better. 

3) Line 48. The authors reported that “…college is a crucial stage of individual psychological and physiological maturity”. What does it mean “college”? Late adolescence? Young adulthood? Please clarify and describe better the mechanisms characterising this developmental stage, considering all study variables. 

4) In “the present study” section, the author states that “Researchers further conceptualise creativity as divergent thinking (fluency, originality, and flexibility) and cohesive thinking”. I would advise the author that divergent thinking is only an index of creative potential. In other words, divergent thinking and creativity are not synonymous. Please address this point in this section and the introduction. 

5) Despite these minor concerns, I believe that the major issue of this paper is the lack of a theoretical basis which can explain the mediating role of psychological adjustment. Maybe, it could be useful to describe throughout the introduction the following relationships: 

  1. Thinking style and Creativity 
  2. Thinking style and psychological adjustment 
  3. Psychological adjustment and creativity 

Also, the authors should explain why they expect the involvement of psychological adjustment in the association between thinking style and creativity. I suggest treating this point carefully since the paper is problematic without a well-grounded theory. Additionally, the author should provide a clear research hypothesis in light of the literature explored previously. 

6) As concerns measures, I suggest authors explain the measure of thinking styles better. In my opinion, it should be more evident that the scale assesses the three dimensions of Chinese thinking style, i.e., Relation, Contradiction, and Change.  

7) In the discussion, the authors should explain their results in light of their hypothesis. Including only the implications is not sufficient. 

8) Finally, the manuscript has some errors and typos (e.g., control variable -> control variables). Please check them. 

In conclusion, based on my reading, the manuscript could be reconsidered after major revision. 

Thank you, and good luck for your research.

English needs improvement.

Author Response

Dear Reviewer,

I would like to express my sincere gratitude for your invaluable guidance. Below, you will find the revisions I have made based on your insightful recommendations. Your feedback has been immensely helpful, and I am deeply appreciative of your support and expertise.

1) The authors based their work on creativity. Nevertheless, I noted that no description and definition of creativity has been provided. I suggest authors include a clear conceptualisation of creativity and its main features. The papers below could be helpful for this aim: 

- Runco, M. A., & Jaeger, G. J. (2012). The standard definition of creativity. Creativity Research Journal, 24(1), 92-96.

- Giancola, M., Palmiero, M., Piccardi, L., & D’Amico, S. (2022). The relationships between cognitive styles and creativity: The role of field dependence-independence on visual creative production. Behavioral Sciences12(7), 212.

Additionally, given that the authors employed the alternative uses task, a clear description of divergent thinking is necessary. The papers below could be useful for this aim:

- Acar, S., & Runco, M. A. (2019). Divergent thinking: New methods, recent research, and extended theory. Psychology of Aesthetics, Creativity, and the Arts, 13(2), 153.

- Giancola, M., Palmiero, M., Bocchi, A., Piccardi, L., Nori, R., & D’Amico, S. (2022). Divergent thinking in Italian elementary school children: The key role of probabilistic reasoning style. Cognitive Processing, 23(4), 637-645.

 Response: Your valuable suggestion is deeply appreciated. In light of this, I have included more precise conceptual definitions within the introduction.

2) The concept of thinking style seems to come out of nowhere. I understand that culture greatly affects people’s way of thinking, but what does represent thinking style?

I advise authors to describe thinking style and then explain the influence of culture. Overall, this description should also be useful for introducing holistic thinking. 

The papers reported below should be useful: 

- Li, L. M. W., Masuda, T., Hamamura, T., & Ishii, K. (2018). Culture and decision making: Influence of analytic versus holistic thinking style on resource allocation in a fort game. Journal of Cross-Cultural Psychology49(7), 1066-1080. 

- Cheng, H., Andrade, H. L., & Yan, Z. (2011). A cross-cultural study of learning behaviours in the classroom: From a thinking style perspective. Educational Psychology31(7), 825-841.

Also, I suggest authors explain thinking style theoretically, reporting a well-established model, allowing the reader to understand the concept of thinking style better. 

Response: I deeply appreciate your valuable suggestion. In response, I have incorporated more specific conceptual definitions into the introduction.

3) Line 48. The authors reported that “…college is a crucial stage of individual psychological and physiological maturity”. What does it mean “college”? Late adolescence? Young adulthood? Please clarify and describe better the mechanisms characterising this developmental stage, considering all study variables. 

Response: I am deeply grateful for your suggestion. The population described in this study comprises university students aged 18-22 years old.

4) In “the present study” section, the author states that “Researchers further conceptualise creativity as divergent thinking (fluency, originality, and flexibility) and cohesive thinking”. I would advise the author that divergent thinking is only an index of creative potential. In other words, divergent thinking and creativity are not synonymous. Please address this point in this section and the introduction.

Response: Your advice has been greatly valued, and as a result, I have revisited the concept of creativity within the article, providing a more detailed discussion regarding its use as an indicator, particularly in terms of divergent thinking. Your insights have significantly enhanced the content, and I deeply appreciate your valuable contributions.

5) Despite these minor concerns, I believe that the major issue of this paper is the lack of a theoretical basis which can explain the mediating role of psychological adjustment. Maybe, it could be useful to describe throughout the introduction the following relationships: 

  1. Thinking style and Creativity 
  2. Thinking style and psychological adjustment 
  3. Psychological adjustment and creativity 

Also, the authors should explain why they expect the involvement of psychological adjustment in the association between thinking style and creativity. I suggest treating this point carefully since the paper is problematic without a well-grounded theory. Additionally, the author should provide a clear research hypothesis in light of the literature explored previously. 

Response: I wish to express my deep gratitude for your valuable input. Not only have I integrated the concept into the introduction, but I have also expanded upon the research hypotheses within "the present study." Your perspectives have played a pivotal role in enhancing the quality of this research.

6) As concerns measures, I suggest authors explain the measure of thinking styles better. In my opinion, it should be more evident that the scale assesses the three dimensions of Chinese thinking style, i.e., Relation, Contradiction, and Change.  

Response: I sincerely appreciate your valuable suggestion. In response, I have incorporated a detailed exposition of the three dimensions of Chinese-style thinking in the introduction. This addition is aimed at providing a clearer explanation of the manipulation in our study.

7) In the discussion, the authors should explain their results in light of their hypothesis. Including only the implications is not sufficient. 

Response: I appreciate your invaluable advice. The outcomes of each study have been integrated into the initial section of the discussion. Your contribution is highly valued.

8) Finally, the manuscript has some errors and typos (e.g., control variable -> control variables). Please check them. 

Response: I am truly grateful for your valuable advice, and I have rectified several mistakes as a result.

I wish to once again convey my heartfelt gratitude and eagerly await your response.

Sincerely,

Huihui Yu.

Round 2

Reviewer 1 Report

Authors did not make any changes in response to concern about ‘psychological maladjustment can lead to negative emotions [14] and even reduce self-efficacy [15], thereby inhibiting the development of creativity.’ This section would benefit from additional explanation and citing appropriate literature explaining the causal link between negative emotions and creativity.

I am not satisfied with the response “Because the value of contradiction is not significant, we did not include it according to academic requirements”. Under-reporting of non-significant results leads to biases (e.g., Mlinaric, A., Horvat, M., & Supak Smolcic, V. (2017) and it is crucial to report all findings, significant or not (Visentin, D. C., Cleary, M., & Hunt, G. E., 2020). Please, make the changes or provide substantiated explanations about specific academic requirements that prevent from reporting full data.

Author Response

Dear Respected Reviewer,

We would like to express out gratitude for your valuable suggestions and feedback. Your input has been invaluable in improving the quality of our work. We would like to share our response to your insightful recommendations:

1. Authors did not make any changes in response to concern about ‘psychological maladjustment can lead to negative emotions [14] and even reduce self-efficacy [15], thereby inhibiting the development of creativity.’ This section would benefit from additional explanation and citing appropriate literature explaining the causal link between negative emotions and creativity.

Response: We greatly appreciate the valuable feedback from the reviewer. We have removed the mentioned statement and made significant revisions to this section, as above content.

2. I am not satisfied with the response “Because the value of contradiction is not significant, we did not include it according to academic requirements”. Under-reporting of non-significant results leads to biases (e.g., Mlinaric, A., Horvat, M., & Supak Smolcic, V. (2017) and it is crucial to report all findings, significant or not (Visentin, D. C., Cleary, M., & Hunt, G. E., 2020). Please, make the changes or provide substantiated explanations about specific academic requirements that prevent from reporting full data.

Response: We also highly appreciate your suggestion; however, following the stepwise regression approach, the variable of contradiction did not enter the model in the end. Therefore, we were unable to report its various statistical indicators.

Thank you for your continued support and collaboration.

Warm regards,

Hou lab.

Reviewer 3 Report

I appreciated that the authors covered my minor concerns. However, I am not convinced about the theory underpinning the model advanced by the authors. In other words, I believe the authors should ground their work by providing a theoretical basis useful to explain the mediating effect of psychological adaptation, explaining why this variable is involved in the association between thinking style and creativity. This mechanism needs to be well justified theoretically. As I have said in the previous round of review, this point needs to be treated very carefully.
Also, in this new version, the authors stated, "This study centered its attention on creativity, particularly emphasizing the indicator of divergent thinking". If so, I wonder why the authors in Study 1 examined "the relationship between thinking styles and creativity through self-report questionnaires".
I suggest, again, to check the English.

There are several errors throughout the manuscript.

Author Response

Dear Respected Reviewer,

We would like to express out gratitude for your valuable suggestions and feedback. Your input has been invaluable in improving the quality of our work. We would like to share our response to your insightful recommendations:

I appreciated that the authors covered my minor concerns. However, I am not convinced about the theory underpinning the model advanced by the authors. In other words, I believe the authors should ground their work by providing a theoretical basis useful to explain the mediating effect of psychological adaptation, explaining why this variable is involved in the association between thinking style and creativity. This mechanism needs to be well justified theoretically. As I have said in the previous round of review, this point needs to be treated very carefully.
Also, in this new version, the authors stated, "This study centered its attention on creativity, particularly emphasizing the indicator of divergent thinking". If so, I wonder why the authors in Study 1 examined "the relationship between thinking styles and creativity through self-report questionnaires".
I suggest, again, to check the English.

Response: We regret that the previous revisions did not provide a better explanation of the theoretical model of the present research. We have revised this section on why psychological maladjustment influences creativity on line 82-93. As follows:

According to the model of factors influencing creativity, both the creative environment and psychological factors play a significant role in impacting the creativity of college students (Lin, 2015). Currently, college students are exposed to complex environments, including academic settings, campus life, and interpersonal relationships. Regardless of the environment’s complexity, fostering creativity requires a democratic and harmonious atmosphere. Harmony, in this context, mainly refers to managing and coordinating various relationships. Psychological harmony and social harmony are aligned (Lin, 2015). Psychological adaptation reflects an individual's response to their surrounding environment and their own psychological state of harmony. Therefore, high psychological adaptation in college students reflects both their positive response to the surrounding environment and their favorable psychological state, providing a foundation for the enhancement of their creativity (Tong et al., 2020).

Reference

Lin, C. (2015). Enhancing adaptive capacity and striving to be creative talents: An address for new college students in school of psychology, Beijing normal university. Studies of Psychology and Behavior, 13(5): 577-584. 

Tong, H., G, Y., & Nan, W. (2020). On the advantages of postgraduates from 2 provinces—A survey and comparison of the quality of postgraduates training in Jiangsu and Zhejiang. Education Research Monthly,(6): 38-44. 

Additionally, we have checked the English.

Once again, we appreciate the time and effort you have dedicated to our work. Your expertise and guidance are truly appreciated, and they have contributed significantly to the enhancement of our project.

Thank you for your continued support and collaboration.

Warm regards,

Hou lab.

Round 3

Reviewer 3 Report

The authors addressed my comments and the paper is now considered suitable for publication.